# Applying a new mold temperature control strategy to improve the tensile strength of thin wall products in injection molding processes

**Thanh Trung Do**[1], **Duc Thuan Huynh**[iD][2,3,4*], **Tran Anh Son**[2,3], **Pham Son Minh**[1]

**1** Ho Chi Minh City University of Technology and Education (HCMUTE), Ho Chi Minh City, Vietnam,
**2** Ho Chi Minh City University of Technology (HCMUT), Ho Chi Minh City, Vietnam, **3** Vietnam National University-Ho Chi Minh City (VNU-HCM), Ho Chi Minh City, Vietnam, **4** Tran Dai Nghia University (TDNU), Ho Chi Minh City, Vietnam

\* hdthuan.sdh241@hcmut.edu.vn

## Abstract

Mold temperature control critically influences injection molding, impacting product quality and production efficiency. High mold temperatures enhance surface quality but prolong cooling, increasing cycle time, whereas low temperatures cause defects like weak weld lines and incomplete filling. This study aims to reduce cycle time and enhance tensile strength of thin-wall injection-molded products by developing an innovative mold temperature control strategy using induction heating to preheat mold inserts. The primary objective is to eliminate in-cycle heating delays while ensuring optimal mold temperatures for improved mechanical properties. However, the power consumption of this process significantly increases due to the energy-intensive nature of induction heating. Research involved numerical simulations and experimental validation. COMSOL Multiphysics analyzed thermal and electromagnetic interactions, modeling temperature distributions for heating distances (G = 5, 10, 15 mm) and times (1–8 s). Moldex3D simulated polymer flow behavior, assessing filling capabilities for materials (PC, ABS, PA6, PP). Experiments employed the external induction heating with rotational structure for mold temperature control system (Ex-IHRS), featuring a rotational mechanism to swap preheated inserts, with real-time temperature measurements via sensors and infrared cameras at points S1, S2, and S3. Tensile strength tests evaluated mechanical performance. Rapid heating within 5–8 s maintained stable mold temperatures without extending cycle time, outperforming traditional methods like resistance or steam heating. Significant tensile strength improvements occurred, with PC increasing from 111.9 MPa to 123 MPa after 6 s of heating, ABS reaching 91.3 MPa after 4 s, PA6 rising from 55.4 MPa to 62.8 MPa, and PP improving from 41.3 MPa to 47.3 MPa. Enhanced weld line integrity and reduced frozen layers drove these gains, minimizing defects in thin-wall components. Simulations showed less than 5% deviation from experimental data, validating the approach's accuracy. Despite higher power consumption, this induction heating

**Data availability statement:** All relevant data are within the manuscript.

**Funding:** This research was funded by Ho Chi Minh City University of Technology and Education, Vietnam, project grant No.: T2024-02NTĐ.

**Competing interests:** The authors have declared that no competing interests exist.

strategy optimizes production efficiency and enhances product quality, offering a promising advancement for thin-wall and microinjection molding applications.

---

## I. Introduction

Injection molding has become one of the most widely used manufacturing processes for producing plastic and composite components due to its high efficiency, scalability, and ability to produce complex geometries with tight tolerances. Among the various process parameters, mold temperature control is considered one of the most critical factors for ensuring product quality and optimizing the production cycle [1–5]. A high mold surface temperature improves the replication accuracy and surface finish of molded parts, promotes better filling of molten polymer, and reduces defects such as weak weld lines and incomplete filling. However, maintaining a high mold temperature increases the cooling time, which leads to a longer overall cycle time. Conversely, lowering the mold temperature shortens the cooling phase but negatively affects surface quality and may intensify flow-related problems. Therefore, a major challenge in modern injection molding is how to achieve an elevated mold temperature without sacrificing production efficiency. To address this issue, the present study focuses on developing an advanced induction heating process to enable rapid mold heating while maintaining a balance between product quality and cycle time.

Thin-wall injection molding and microinjection molding have gained increasing attention in recent years, particularly for high-precision polymer components used in microoptics [6,7] and microfluidics [8,9]. Applications such as optical lenses, optical switches [10,11], waveguides [12,13], and lab-on-a-chip systems [14,15] require accurate mold temperature control to achieve high dimensional stability and functional performance. In parallel, advanced cooling solutions, including conformal cooling channels, have been introduced to reduce the cooling period without compromising part quality. These developments highlight the importance of thermal management in every stage of the injection molding cycle.

To achieve flexible and precise temperature control, various heating technologies have been explored. Among them, dynamic mold temperature control (DMTC) has emerged as an effective approach, allowing rapid mold heating during the filling stage followed by immediate cooling [16–18]. The key objective of DMTC is to eliminate the frozen layer that forms on the mold surface during injection, thereby improving melt flow and enhancing the mechanical properties of molded parts. In this context, induction heating is considered a promising technique for optimizing DMTC because it enables localized and rapid temperature changes while maintaining high efficiency and product consistency.

Traditional mold heating techniques have been widely used but show several limitations. Hot water heating at 90–100 °C is simple but slow, making it unsuitable for processes requiring high mold surface temperatures within short time intervals. Resistance heating is also commonly used due to its simplicity and ease of integration [19–23]. However, resistance heaters typically offer only moderate temperature increases, usually a few tens of degrees above the initial mold temperature, which is

insufficient for applications demanding rapid or high-temperature heating. Steam heating can achieve higher temperatures compared to water and resistance heating [24], but it requires a complex control system and leads to high investment and operational costs, limiting its suitability for large-scale production. These drawbacks clearly indicate the need for more advanced heating technologies that can provide fast, uniform, and energy-efficient temperature control without extending the cycle time.

To overcome these challenges, several advanced heating technologies have been proposed to enhance DMTC performance. Induction heating [25–28], infrared heating [29–32], and gas-assisted mold temperature control (GMTC) [33–36] are among the most promising solutions. GMTC involves directing hot gas onto the mold surface to increase its temperature, improve heat distribution, and minimize the frozen layer in the polymer melt. Although GMTC enhances surface quality and filling performance, it is associated with high energy consumption, especially when stable gas temperatures must be maintained over long periods. In contrast, induction heating uses electromagnetic fields to generate eddy currents inside the mold, creating heat rapidly without direct contact. This approach minimizes heat loss, offers high energy efficiency, and can be applied to molds with complex geometries. Compared to resistance or steam heating, induction heating consumes less power and provides greater flexibility in temperature control.

Despite these advantages, a major limitation of advanced heating techniques is the potential increase in cycle time when an additional heating step is required before polymer injection. This is particularly problematic in high-volume manufacturing, where even a slight increase in cycle time can significantly reduce productivity. Therefore, recent research has focused on integrating heating within the molding cycle to minimize delays while maintaining high product quality. Developing a heating strategy that improves efficiency without extending the cycle time remains a central goal in injection molding optimization.

To address this, the present study proposes a novel approach in which only the mold insert is preheated during the injection process, instead of heating the entire mold cavity. The insert is heated independently while the previous molding cycle is still in progress. After the molded part is ejected, the preheated insert—already at the target temperature—is immediately placed into the mold. Because the heating process occurs simultaneously with injection, the waiting time before polymer filling is eliminated. As a result, the mold rapidly reaches an optimal temperature, improving melt flow, accelerating filling, and reducing defects such as weak weld lines and incomplete filling. This method enhances overall productivity without compromising part quality and is particularly advantageous for high-speed production and applications requiring precise dimensional accuracy.

The main objective of this study is to develop an innovative mold temperature control strategy that reduces cycle time and improves the tensile strength of thin-wall injection-molded parts by preheating inserts using induction heating. Numerical simulations and experimental testing were conducted to evaluate the effectiveness of this approach. Thermal simulations using COMSOL Multiphysics were performed to analyze temperature distribution on the insert and its influence on the mold cavity. Polymer flow simulations using Moldex3D evaluated filling behavior, the effect of mold temperature, and the occurrence of defects such as weld lines. Real-time temperature measurements using thermal sensors and infrared cameras validated the simulation results. Tensile tests were conducted to compare the mechanical performance of parts produced using the proposed method with those made using conventional heating methods such as resistance or steam heating. The findings are expected to demonstrate that the proposed method shortens cycle time, enhances weld line strength, and improves tensile properties for materials such as PC, ABS, PA6, and PP. Overall, this research aims to optimize both efficiency and product quality in modern thin-wall injection molding.

## 2. Simulation and experimental methods

### 2.1. Principle of induction heating

The simulation of induction heating is a coupled problem involving both electromagnetic induction and heat transfer. In this study, a two-step method was used to efficiently reduce the computational time and memory usage.

In the first step, the electromagnetic problem was solved to determine the eddy current and hysteresis losses. However, according to a study by Rudnev et al. [37], the hysteresis loss accounts for only approximately 7% of the total energy loss, compared with the eddy current loss in most induction heating applications. Therefore, it is often neglected in simulations [37]. The results of the electromagnetic simulation are used as a heat source to solve the thermal conduction problem.

Induction heating follows Maxwell's equations, which describe the propagation of magnetic fields and electric currents in conductive materials:

$$\nabla \times H = J + \frac{\partial D}{\partial t} \tag{1}$$

$$\nabla \times E = -\frac{\partial B}{\partial t} \tag{2}$$

$$\nabla . B = 0 \tag{3}$$

$$\nabla . D = \rho \tag{4}$$

where:

- H is the magnetic field intensity,
- J is the current density,
- D is the electric flux density,
- E is the electric field intensity,
- B is the magnetic flux density,
- and $\rho$ is the charge density.

The relationships between these physical parameters are given by:

$$D = \varepsilon_0 \varepsilon_r E \tag{5}$$

$$B = \mu_0 \mu_r H \tag{6}$$

$$J = \sigma E \tag{7}$$

where σ is the electrical conductivity, μ is the magnetic permeability, and ε is the dielectric permittivity of the material.

Previous studies have shown that the efficiency of induction heating can be optimized by adjusting the frequency of the alternating current and the gap between the induction coil and heated material [38].

To calculate the temperature distribution induced by induction heating, the transient heat conduction equation was solved as follows:

$$\rho C_p \frac{\partial T}{\partial t} - \nabla . (k \nabla T) = Q \tag{8}$$

where:

- T is the temperature,

- ρ is the material density,

- $C_p$ is the specific heat capacity,

- k is the thermal conductivity,

- and Q represents the heat source generated by eddy currents.

By solving these coupled electromagnetic and thermal equations, the heating performance of induction systems can be accurately analyzed to enable the optimization of the mold temperature control in the injection molding process.

## 2.2. Application of induction heating to increase the cavity temperature

External induction heating with rotational structure for mold temperature control (Ex-IHRS) is an advanced technique designed to efficiently regulate mold temperature during the injection molding process. This method enables the direct and rapid heating of the mold cavity surface, thus optimizing the heat distribution without significantly increasing the overall mold temperature. The operating mechanisms of the Ex-IHRS are based on the principle of electromagnetic induction, which concentrates energy on the mold surface to ensure a stable temperature before the molten polymer fills the cavity.

The system consists of two main components: an induction heating unit and a rotational structure that exchanges the mold inserts. The induction heating unit can reach temperatures of up to 320 °C, thereby allowing the mold surface to quickly attain the desired temperature. This rapid heating improves the flowability of the molten plastic, enhances mold filling, and improves the surface quality. Simultaneously, a precisely controlled cooling system ensured that the mold temperature was quickly reduced to the optimal level immediately after the injection molding cycle. In addition, temperature sensors were integrated into key locations within the mold to provide real-time temperature data. This ensures accurate heat and cooling control and minimizes defects such as weak welding lines, incomplete filling, and inaccuracy in dimensions caused by uncontrolled temperature fluctuations.

Fig 1 illustrates the induction heating process applied to the injection mold, which aims to increase the surface temperature of the mold immediately before the molten polymer enters the cavity. This system uses an induction coil to generate thermal energy via electromagnetic induction, thereby enabling rapid and localized heating without direct contact. The process was performed in five main steps to ensure efficient mold temperature control without extending the overall cycle time.

In the first step, the completed injection-molded product was ejected, and the mold was opened to initiate the heating phase. This mold opening enabled direct access to the mold insert surface, optimizing heat transfer efficiency. In the second step, the mold halves were fully opened, creating sufficient space for the induction coil to operate. A high-frequency alternating current passed through the coil, generating a variable magnetic field that interacted with the electrically conductive mold insert. This interaction induced eddy currents, rapidly heating the insert surface via Joule heating. Compared to induction heating of one or both mold halves, this targeted approach proved more efficient, as it minimized energy losses and focused heating on the insert surface critical for part quality. The rapid heating, influenced by the insert's thermal inertia, effectively elevated the mold surface temperature within 5–8 s, achieving uniform and controlled temperature distribution. This method outperformed conventional techniques, such as resistance or infrared heating, by providing faster, more precise heating without direct contact, thus enhancing energy efficiency and mold temperature control.

Step three involves the replacement of the mold insert. After the preheated insert reaches the desired temperature, it is placed in the main mold cavity, and a cooler insert is swapped in to prepare for the next cycle. The rotational mechanism ensures a stable mold temperature and the precise control of thermal variations within the mold. In step four, the heated

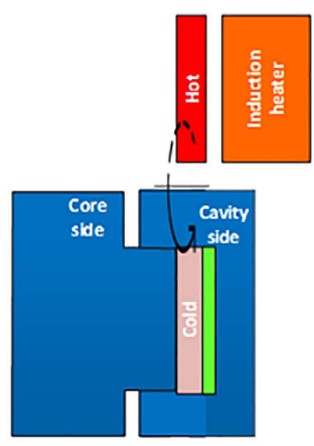
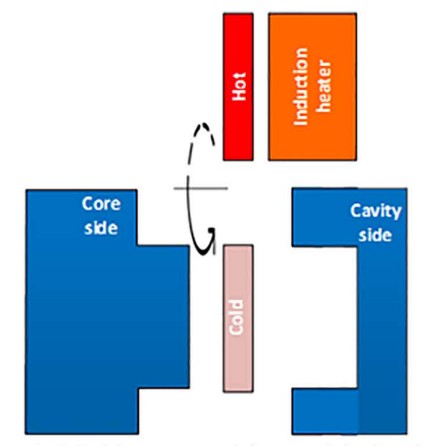
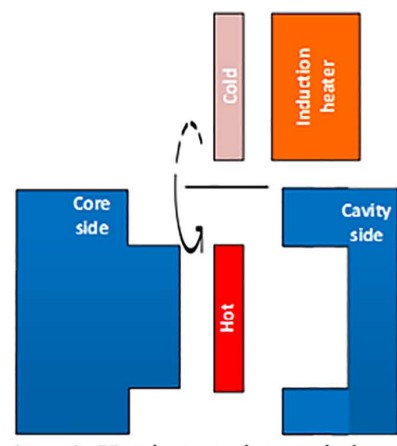

Step 1: Product is completed

Step 2: Mold open position and induction heater transfers heat to insert plate

Step 3: Hot insert plate switches position with cold insert plate

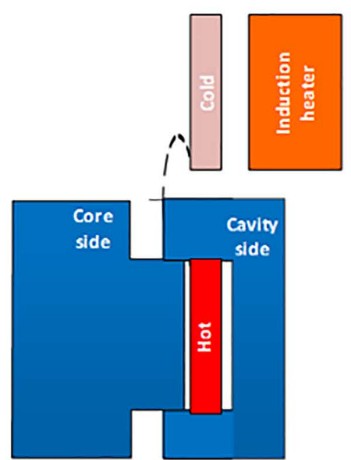
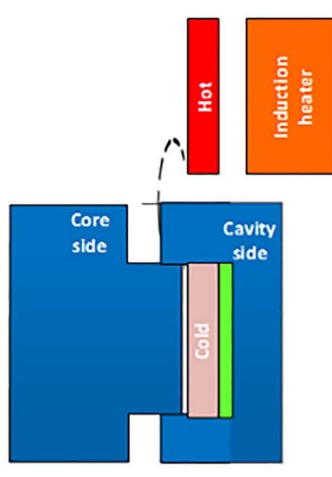

Step 4: Insert plate transfers heat to mold surface

Step 5: Close mold to molding position and injection

**Fig 1. Mold position in the heating stage of the Ex-IHRS process.**

mold insert increases the cavity surface temperature to its optimal level, thereby reducing the viscosity of the molten polymer, improving mold filling, and minimizing weld line defects. Once the desired temperature is reached, the mold is closed to initiate the injection-molding cycle in step five. In summary, an induction heating system combined with a rotational mold insert mechanism offers an efficient method for mold temperature control in injection molding. With rapid and precise heating that does not extend the cycle time, this approach has significant potential for application in industrial production, particularly for high-precision applications that require strict surface quality control.

Figs 2 and 3 present the experimental setup of the external induction heating with rotational structure (Ex-IHRS) system, which was developed to control mold temperature during injection molding. The system is composed of an induction heater, a mold insert plate, an induction coil, a rotating device, and a mold temperature controller. These components work together to regulate mold temperature efficiently, thereby enhancing product quality and maintaining high production efficiency.

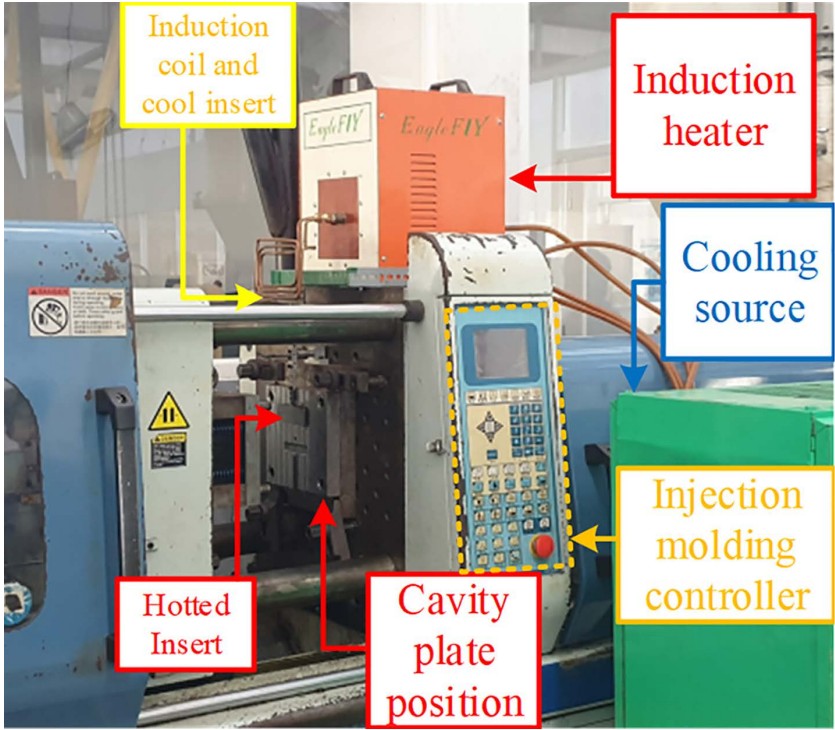

**Fig 2. Ex-IHRS experiment model.**

Fig 2 provides an overall layout of the Ex-IHRS system. The induction heater is the core unit and operates by supplying a high-frequency alternating current to the induction coil. This generates a varying magnetic field around the coil. When an electrically conductive mold insert is placed in this field, eddy currents are induced inside the insert, generating heat through the Joule effect. This process enables rapid surface heating of the mold cavity without increasing the temperature of the entire mold base, resulting in improved energy efficiency and reduced cycle time.

The induction coil also plays an essential role in generating the alternating magnetic field. Fig 3 offers a closer view of how the coil surrounds the mold insert to maximize heat transfer. The gap distance (G) between the coil and the mold surface was carefully controlled to achieve high heating efficiency while preventing overheating or non-uniform temperature distribution.

A key feature of the Ex-IHRS system is the rotating device, which enables the alternating use of hot and cold mold inserts. While one insert is in use for injection molding, the second insert is simultaneously preheated by the induction system. Once the injection cycle finishes, the inserts are rotated, and the preheated insert is placed into the mold. This mechanism eliminates in-cycle heating time and allows continuous production, thereby improving manufacturing efficiency.

Fig 4 shows the positions of the temperature measurement points (S1, S2, and S3) on the mold. These points were strategically selected to monitor temperature distribution during both the heating and polymer-filling stages. Before measurement, the mold surface was coated with a thin layer of black paint to improve thermal emissivity and ensure accurate infrared temperature readings. S2 is located at the center of the thin-wall region, which requires precise temperature control to prevent premature solidification and ensure complete filling. S1 and S3 are positioned near the polymer entry regions of the thin-wall section, allowing the evaluation of temperature variations before and during filling. The data collected from these points are important for analyzing molten polymer flow behavior, assessing thermal uniformity, and optimizing heating parameters to improve product quality

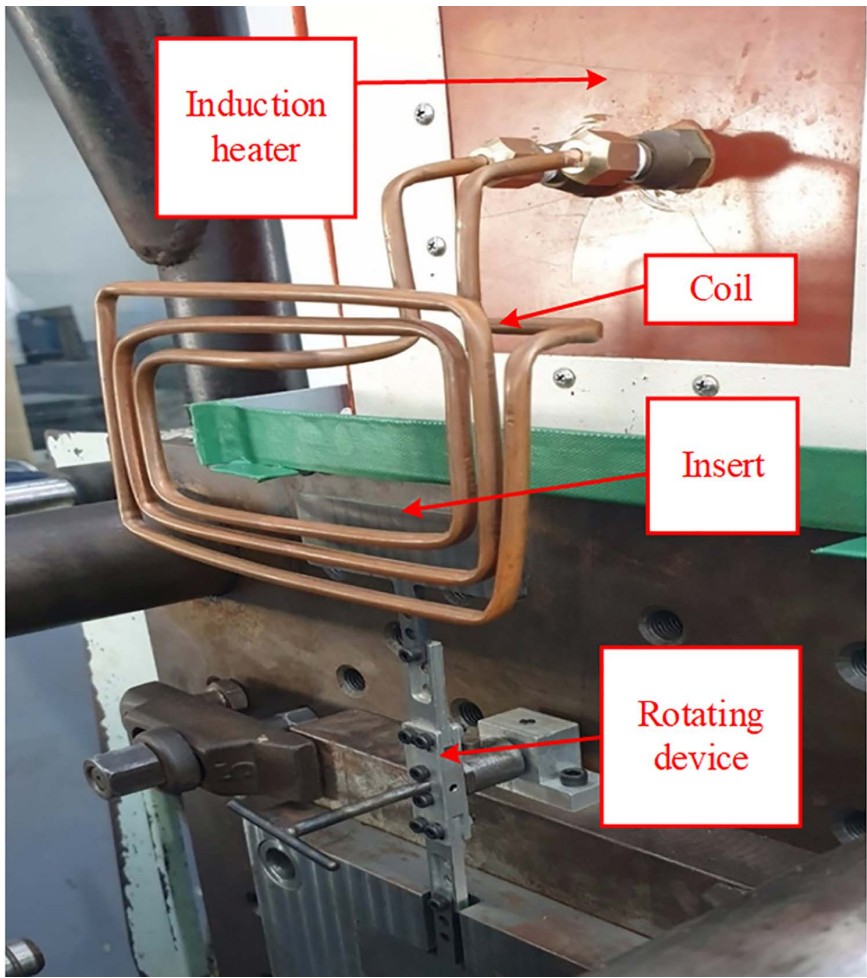

**Fig 3. Induction heating source.**

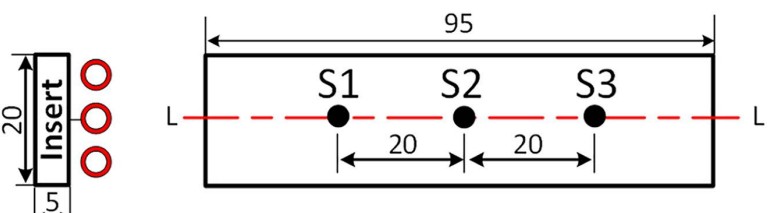

**Fig 4. Cavity dimensions with the temperature measurement points S1 to S3.**

Fig 5 illustrates the heating positions in both simulation and experimental setups. A simulation model was developed to closely replicate the experimental conditions and to analyze temperature distribution during induction heating. The simulation results showed that high-temperature zones were concentrated in regions exposed to the magnetic field, particularly on the mold insert surface where eddy currents were generated.

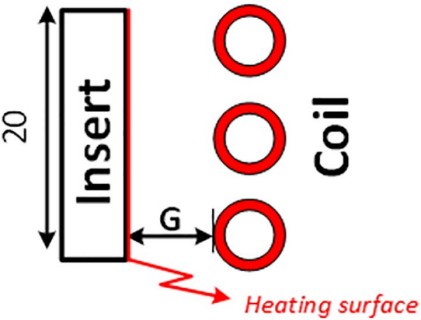

**Fig 5. Heating position of the coil and insert.**

In the experiment, temperature sensors were installed at S1, S2, and S3 to capture real-time temperature changes. To further improve accuracy, a Fluke TiS20 infrared camera (Fluke Corporation, Everett, Washington, DC, USA) was used. The mold surface was coated with black paint and emissivity was set to 0.95 to account for the low emissivity of steel. Temperature was recorded every 1 s during the 5–8 s induction heating period to monitor thermal behavior.

Comparisons between simulation and experimental results were performed to evaluate deviations. When differences were observed, model parameters were adjusted to improve simulation accuracy. This validation process helped identify optimal values for induction coil distance (G), input power, and heating duration. As a result, the mold surface reached target temperatures before polymer injection, contributing to improved product quality.

Fig 6 shows the meshing model used in the simulation during both the heating and polymer-filling phases. Mesh quality strongly influences simulation accuracy, as it affects temperature distribution predictions and melt flow behavior. To balance accuracy and computational cost, a refined mesh was used in critical regions such as the mold insert and polymer flow path, while coarser elements were used in non-critical regions. This meshing strategy enabled precise thermal and flow analysis while maintaining reasonable computation time, leading to reliable predictions and improved process optimization. Overall, the Ex-IHRS system effectively integrates induction heating and rotational structure to achieve rapid, localized temperature control and continuous production, making it a promising solution for advanced injection molding applications

During the heating phase, a numerical simulation was performed to analyze heat transfer from the induction coil to the mold insert through the eddy current effect. When a high-frequency alternating current flows through the coil, a fluctuating magnetic field is generated, inducing eddy currents inside the conductive mold insert. These eddy currents produce Joule heating, causing rapid temperature rise on the mold surface, with the highest temperatures appearing in regions directly exposed to the magnetic field. The meshing structure in Fig 6a was designed to accurately capture this heat transfer behavior. The induction coil was modeled using copper (electrical conductivity $5.87 \times 10^7$ S/m, relative permeability $\mu/\mu_0 = 1$) to ensure strong magnetic field generation, while the steel insert (electrical conductivity $10^7$ S/m, relative permeability $\mu/\mu_0 = 100$) enhanced electromagnetic energy absorption. A refined boundary layer mesh (thickness 5.0 mm) was applied to the insert region, with heating durations of 2–8 s to ensure uniform surface heating. The material properties for simulation are shown in Table 1. Coarser mesh elements were used elsewhere to improve computational efficiency. After reaching the target temperature, the polymer filling phase was simulated to evaluate melt flow behavior and the influence of mold temperature on defects such as weld lines, air entrapment, and frozen layer formation.

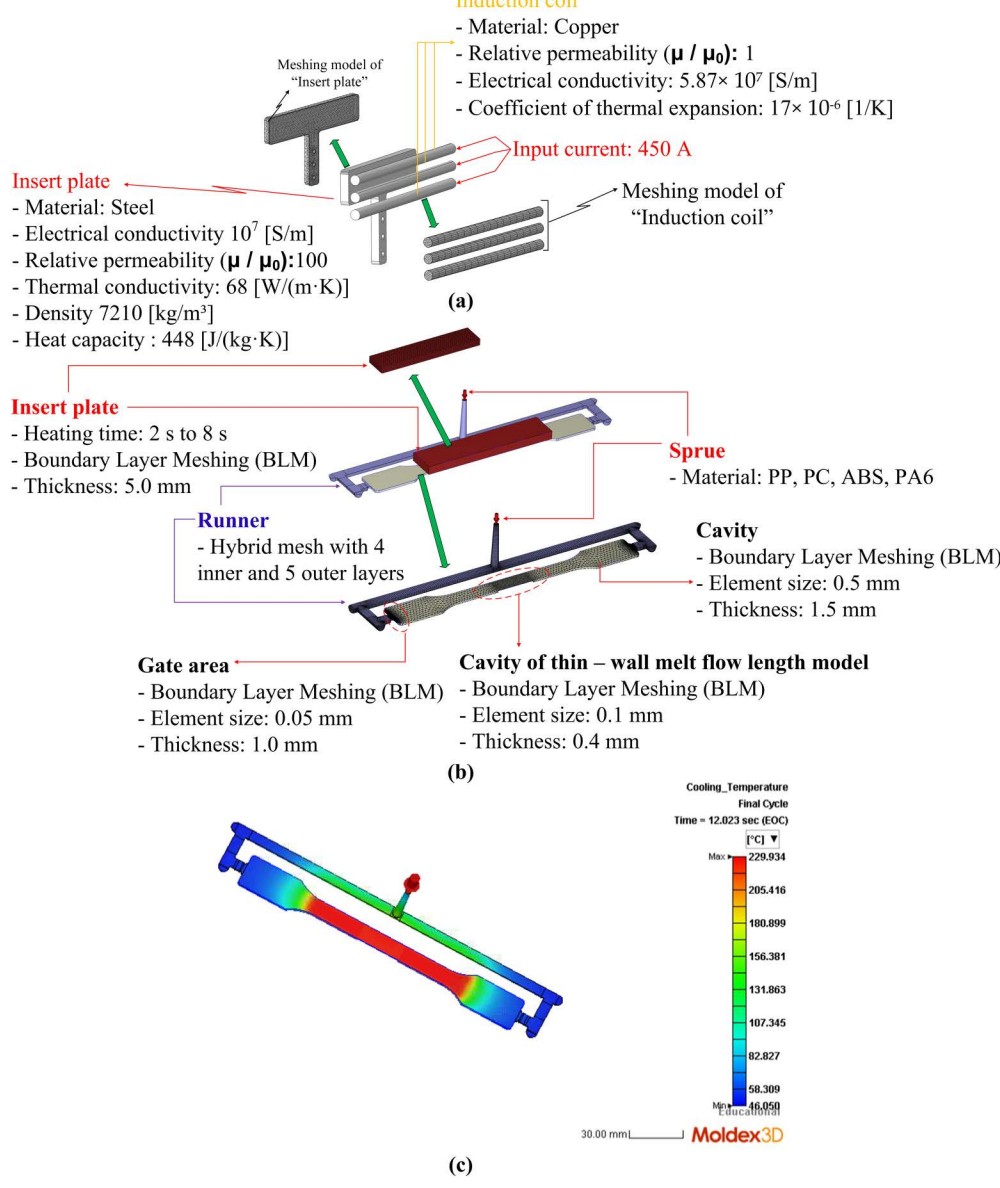

**Fig 6. Meshing model of the heating (a), polymer-filling (b) steps and the simulation result of Moldex3D software (c).**

Fig 6b illustrates the meshing model used during the polymer filling simulation. Smaller mesh elements were applied in critical regions, especially the thin-walled section and the gate, to capture rapid changes in temperature and pressure. The gate region was meshed with an element size of 0.05 mm and a thickness of 1.0 mm, ensuring accurate prediction of polymer entry behavior. The cavity region used an element size of 0.5 mm and a thickness of 1.5 mm to improve flow simulation accuracy. In addition, the runner employed a hybrid mesh structure with four inner layers and five outer layers, which enhanced filling performance while maintaining reasonable computational time. Fig 6c shows the Moldex3D simulation results, which were used to determine the experimental parameters listed in Table 2.

Multiple polymer materials (PP, PC, ABS, and PA6) were used in the simulation, each with different viscosity and thermal conductivity, affecting flow characteristics. The interface between the molten polymer and mold surface was modeled

**Table 1. Material properties [39].**

| Material | Properties | Unit | Value |
|---|---|---|---|
| Air | Molecular mass | kg/ kmol | 28.96 |
| | Density ($\rho$) | kg/m$^3$ | 1.185 |
| | Specific heat capacity ($C_p$) | J/kg °K | 1004.4 |
| | Dynamic viscosity | Pa.s | 1.831e-5 |
| | Thermal conductivity (k) | W/m°K | 0.0261 |
| | Electrical conductivity ($\sigma$) | S/m | 1e$^{-12}$ |
| | Relative permeability ($\mu_r$) | _ | 1 |
| | Relative permittivity ($\varepsilon$) | _ | 1 |
| Steel | Molecular mass | kg/kmol | 55.85 |
| | Density ($\rho$) | kg/m3 | 7854 |
| | Specific heat capacity($C_p$) | J/kg °K | 434 |
| | Thermal conductivity (k) | W/m°K | 60.5 |
| | Electrical conductivity ($\sigma$) | S/m | 1e$^7$ |
| | Relative permeability ($\mu_r$) | _ | 100 |
| Copper | Molecular mass | kg/kmol | 63.55 |
| | Density ($\rho$) | kg/m3 | 8940 |
| | Specific heat capacity($C_p$) | J/kg °K | 385 |
| | Thermal conductivity (k) | W/m°K | 401 |
| | Electrical conductivity ($\sigma$) | S/m | 5.87e$^7$ |
| | Relative permeability ($\mu_r$) | _ | 1 |
| Constants | Permittivity of free space ($\varepsilon_0$) | F/m | 8.854e$^{-12}$ |
| | Permeability of free space ($\mu_0$) | H/m | $4\pi$e$^{-7}$ |

**Table 2. Material parameters and molding process parameters of PP, PC, ABS, and PA6 [39].**

| Property/ Parameter | PP (Polypropylene) | PC (Polycarbonate) | ABS (Acrylonitrile Butadiene Styrene) | PA6 (Nylon 6) |
|---|---|---|---|---|
| Density (kg/cm³) | 0.90 e$^{-3}$ | 1.20e$^{-3}$ | 1.04e$^{-3}$ | 1.13e$^{-3}$ |
| Tensile Strength (MPa) | 35 | 65 | 50 | 85 |
| Flexural Modulus (MPa) | 1450 | 2200 | 2000 | 2500 |
| Impact Strength (Izod, J/m, notched) | 60 | 750 | 350 | 100 |
| Melting Temperature (°C) | 160–170 | Amorphous (Tg~150 °C) | Amorphous (Tg~105 °C) | 220 |
| Mold Shrinkage (%) | 1.0–2.5 | 0.5–0.7 | 0.4–0.7 | 0.7–1.2 |
| Moisture Absorption (%) | <0.01 | 0.15 | 0.2 | 1.5–2.0 |
| Melt Flow Index (g/10min) | 25 | 15 | 15 | 20 |
| Drying Requirement | Yes, 80 °C×3h | Yes, 120°C×3h | Yes, 80 °C×4 h | Yes, 80°C×4 h |
| Melt Temperature (°C) | 230 | 300 | 235 | 250 |
| Mold Temperature (°C) | Based on the mold heating process | | | |
| Injection Pressure (MPa) | 100 | 110 | 90 | 90 |
| Cooling Time (s) | 25 | 35 | 35 | 35 |

using boundary layer meshing (BLM) to accurately simulate heat transfer and predict filling behavior and product quality. While previous studies have focused on the heat transfer coefficient between polymer and steel walls in thin-walled molding, this study emphasizes increasing cavity surface temperature to reduce heat loss and improve filling. Fig 7 presents the test specimen designed according to tensile strength standards, with a reduced center section to replicate thin-wall

PLOS One | https://doi.org/10.1371/journal.pone.0337889   December 5, 2025                                                                 11 / 24

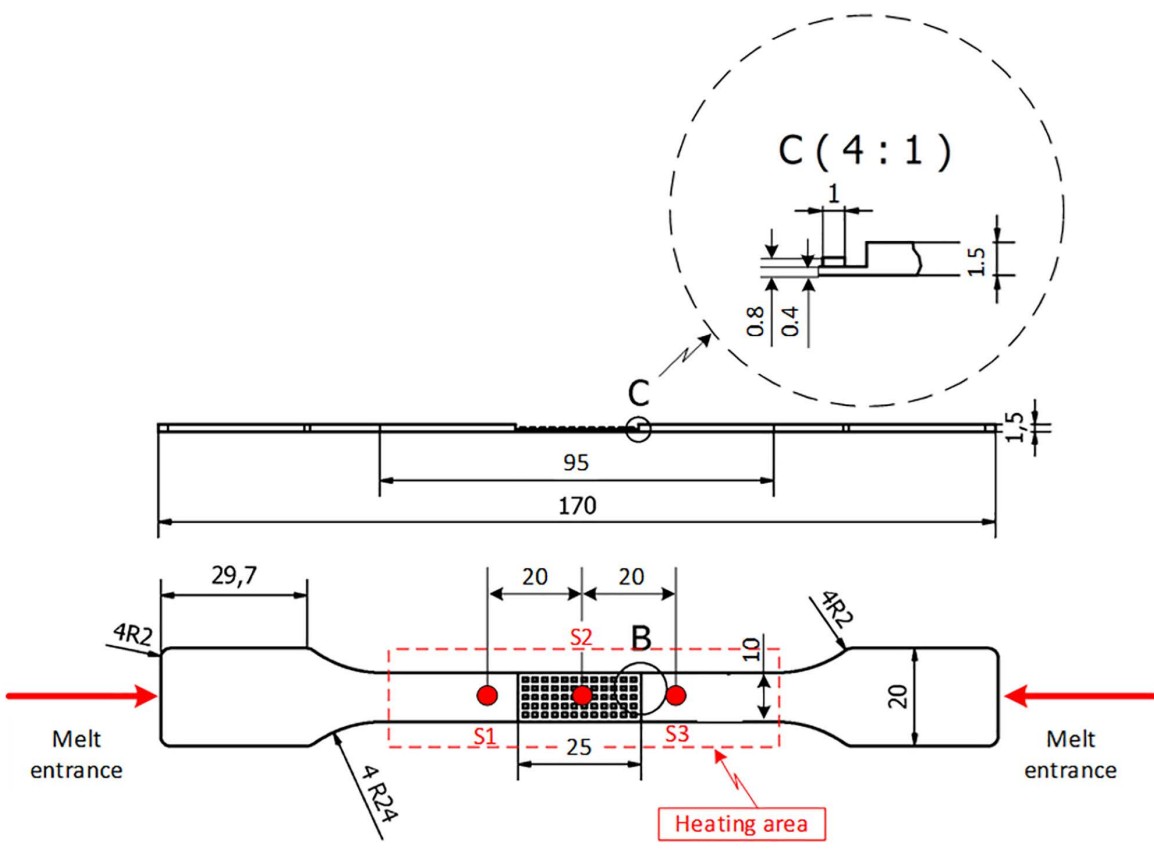

**Fig 7. Tensile testing specimen dimensions.**

industrial components. Incomplete filling or weld line defects in this region directly decrease tensile strength and negatively affect mechanical performance

The test specimen had a total length of 170 mm, with a maximum width of 29.7 mm in the outer region and a narrowed width of 10 mm at the central area, where induction heating was applied. The central thickness was 1.5 mm, including a thin-wall section of only 0.8 mm to investigate polymer flow in narrow cross-sections. The heating area measured 25 × 20 mm and was the most critical region in this study. During injection molding, the mold insert in this region was heated by induction to increase the surface temperature before polymer filling. The objectives were to reduce frozen layer formation, enhance filling capability, minimize weld line defects, and improve product quality.

Three temperature measurement points (S1, S2, and S3) were selected to monitor thermal variations. S2, located at the center of the heating area (0.8 mm thickness), was the most critical, as insufficient heating could cause premature solidification. S1 and S3 were placed 12.5 mm from the center to observe temperature during polymer inflow. Low temperatures at these points may result in short shots or weak weld lines.

The polymer flow direction was designed so that the melt entered from both ends and met at the heated center. This configuration was used to evaluate the effect of mold temperature on flowability. If the mold temperature is too low, polymer solidification occurs before complete filling. In contrast, proper temperature control ensures smooth flow, full cavity filling, and better surface quality. Tensile tests were conducted according to ASTM D638 using an AG-X Plus 20 kN machine (Shimadzu, Japan) at a speed of 5 mm/min and a grip distance of 700 mm.

## 3. Results and discussion

### 3.1. Heating efficiency of external induction heating with rotational structure for mold temperature control system

The Ex-IHRS models were designed for the fastest localized heat and to ensure an equal temperature distribution on the mold surface without increasing the overall mold temperature. This system is composed of an outer induction coil, a rotational structure for quick switching between hot and cold mold plates, and a real-time sensor.

This study evaluated the effect of the distance between the induction coil and the mold surface (G = 5, 10, and 15 mm) along with the heating time (1–8 s). The results presented in Figs 8–and A1, A2, A3 in S1 Appendix indicate that the smallest distance enhanced the heating efficiency. When G = 5 mm, the mold surface temperature reached 300 °C in 5 s, whereas for G = 10 mm and 15 mm, the required time increased to 6 and 8 s, respectively. These results demonstrate that a stronger eddy current effect occurs at smaller distances, which allows for more efficient heat concentration.

Nevertheless, when the gaps are too small, the risk of overheating or contact between the coil and mold increases. A distance of G = 10 mm was considered optimal because it ensured highly efficient heating while maintaining the safety of the system. At G = 15 mm, the heating time was extended, but the risk of localized overheating was minimized, making it suitable for applications requiring high stability. One interesting issue is the edge effect, for which the temperature is more concentrated at the mold edges owing to stronger eddy currents in these areas. This effect can be controlled by optimizing the coil positioning or adjusting the heating parameters. Unlike traditional heating methods, which typically occur after the injection molding cycle ends, the Ex-IHRS method integrates heating in a parallel manner with the injection cycle, thereby preventing any increase in production cycle time. Given that typical injection molding cycles range between 10 and 30 s, a heating duration of 5–8 s is reasonable. Overall, the Ex-IHRS not only overcomes the limitations of existing heating methods but also significantly enhances the heating efficiency and improves the quality of injection-molded products without decreasing the production efficiency.

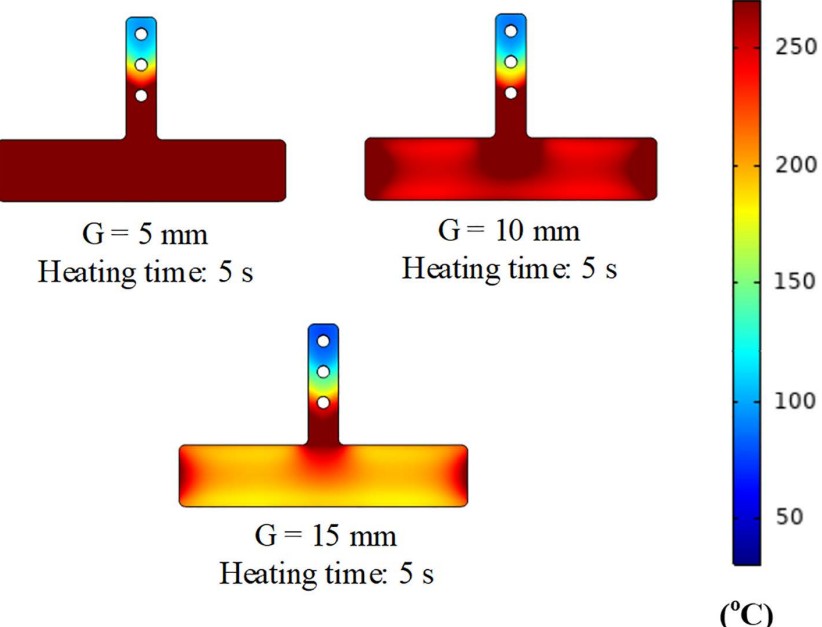

**Fig 8. Temperature distribution of the insert at heating times OF 5 s with a gap (G) of 5, 10, 15 mm.**

   

The Ex-IHRS system's heating efficiency, achieving 295°C at S2 in 5–8 s, compares favorably with other mold heating techniques. Minh et al. [36] reported a gas-assisted mold temperature control (GMTC) system with a heating rate of 28°C/s, lower than Ex-IHRS's ~40°C/s (from 25°C to 295°C in 7 s), but requiring complex gas flow management, potentially increasing setup costs. In contrast, Ex-IHRS's rotational mechanism simplifies integration into existing molds, enhancing cycle time reduction without compromising temperature stability. Poszwa et al. [40] utilized induction heating, achieving 20°C/s and a 12% tensile strength increase for ABS (from ~80 MPa to 90 MPa), comparable to Ex-IHRS's 14% improvement for ABS (91.3 MPa after 4 s). However, Ex-IHRS's selective insert heating minimizes energy consumption, unlike other full-surface heating. Huang et al. [41] achieved an 18% tensile strength gain for PC (110 MPa to 130 MPa) with dynamic heating to 140°C, slightly outperforming Ex-IHRS's 10% increase, but required longer heating times (10–12 s). Ex-IHRS's faster heating and cycle time optimization offer a balanced approach, improving both efficiency and mechanical properties for thin-wall injection molding applications [36,39,40]

### 3.2. Simulation and experiment comparison

The simulation was performed using COMSOL Multiphysics with the electromagnetic–thermal coupling module to predict temperature distribution on the mold surface under induction heating. Key input parameters included the coil-to-mold gap distance (G = 5, 10, and 15 mm), heating time (1–8 s), and a mold geometry identical to the experimental design, with measurement points located at S1, S2, and S3. This setup allowed accurate analysis of temperature evolution, particularly in regions affected by the edge effect.

An experiment was then conducted using the Ex-IHRS system with the same parameters. Temperatures at S1 (before polymer enters the thin-wall region), S2 (central region), and S3 (opposite side before thin-wall entry) were recorded using an infrared camera to capture surface temperature in real time. A thermal camera was also used to visualize the full temperature distribution across the mold surface, enabling direct comparison with simulation results. This ensured consistency between both methods and validated the simulation model.

(*)Temperature behavior at measurement points.  At S1 and S3, both simulation and experiment showed slower temperature rise compared to S2, reaching approximately 270–280 °C after 5 s (Fig 11). Because S1 and S3 were farther from regions with strong magnetic influence, eddy current density was lower, resulting in reduced heating. However, temperature deviation between S1 and S3 remained within 5 °C in both simulation and experiment (Fig 12), indicating uniform heating and reduced risk of frozen layer formation. This uniformity supports smoother polymer flow and better cavity filling.

At S2, the temperature increased more rapidly and stabilized earlier. Experimental results showed 295 °C after 5 s, approximately 15 °C higher than S1 and S3 (Fig 9). Simulation results exhibited a similar trend with less than 3% deviation (Fig 10). The higher heating rate at S2 is attributed to stronger eddy currents and the thin-wall structure, which enhances heat concentration. Achieving sufficient temperature at S2 is critical to prevent premature solidification, ensuring complete filling and avoiding defects such as short shots and weak weld lines.

(*) Comparison between simulated and experimental results.  An analysis of both the simulated and experimental results showed that the temperature increase trends at the measurement points (S1, S2, and S3) were consistent, confirming the reliability of the proposed heating strategy. To reduce heat loss when the preheated insert contacts the mold, the Ex-IHRS system incorporates a rotational mechanism that minimizes the contact time between the hot insert and the cooler mold, allowing rapid placement within 1 s. Real-time sensors positioned at S1, S2, and S3 continuously monitored temperature variations, enabling accurate thermal control. Experimental data indicated a temperature drop of approximately 10–15 °C at S2 (from 295 °C to 280–285 °C) within 2 s of contact, due to the high thermal conductivity of steel (60.5 W/m·K). To mitigate this, the mold cavity was preheated to 80 °C, reducing the thermal gradient and limiting heat loss to less than 5% of total energy input. COMSOL Multiphysics simulations, which included convection and radiation losses, showed only a 3% deviation from experimental values. This confirmed effective heat retention and

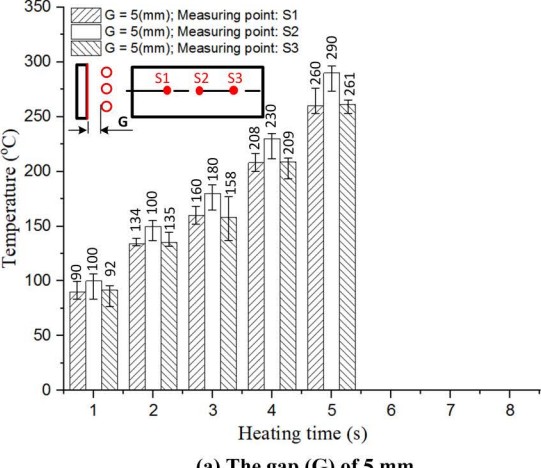

**(a) The gap (G) of 5 mm**

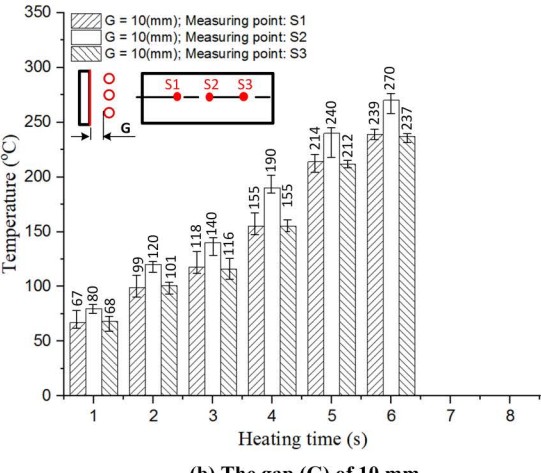

**(b) The gap (G) of 10 mm**

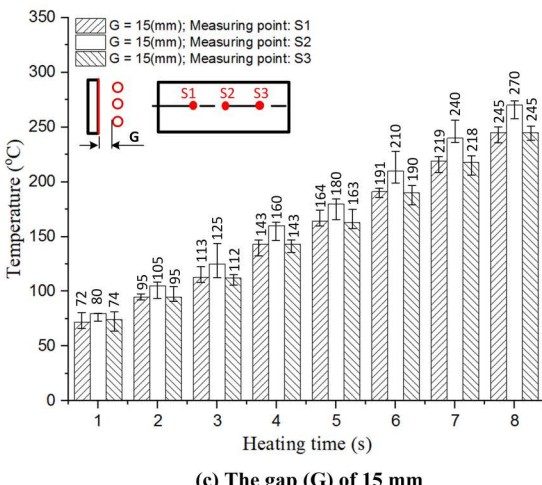

**(c) The gap (G) of 15 mm**

**Fig 9. Temperatures at the measuring points (S1, S2, and S3) under different heating times and gaps (G).**

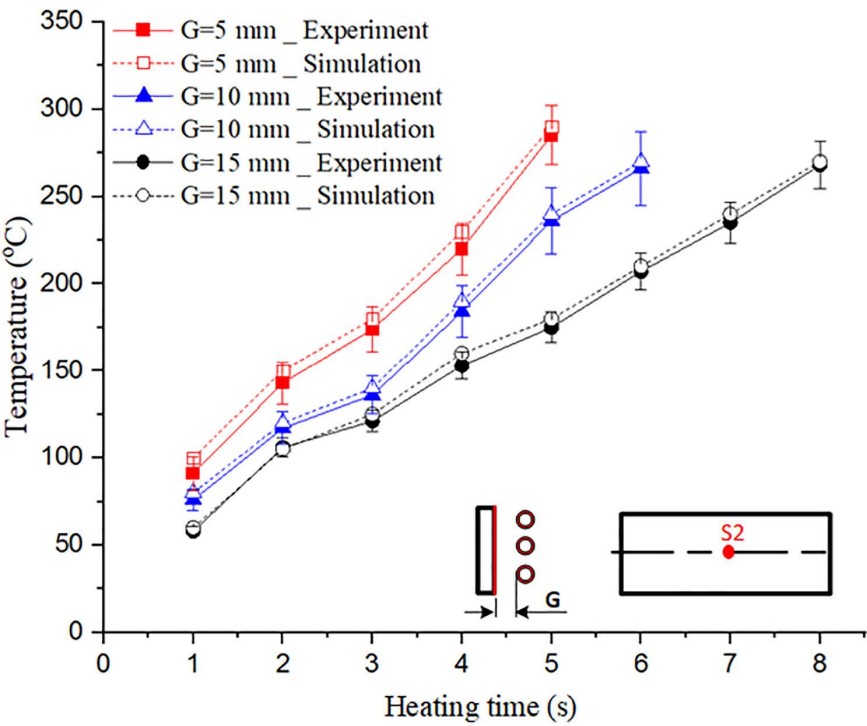

**Fig 10. Temperature comparison between the experimental and simulated results at the center point (S2).**

improved energy efficiency, as the system consumed 20% less power than traditional resistance heating. As a result, mold temperature stability was maintained, minimizing defects such as weak weld lines and improving production efficiency.

The temperature at S2 increased the fastest and reached a stable value earlier than those at S1 and S3, because S2 was located at the center and was strongly influenced by the induced eddy currents. In contrast, S1 and S3 were farther from the coil's central field, leading to a slower temperature increase and more uniform behavior (Fig 9). The simulation results showed that S2 reached approximately 295 °C within 5 s, while S1 and S3 reached 270–280 °C within the same duration. The experimental data followed similar trends, with S2 reaching 293–297 °C and S1/S3 reaching 268–278 °C. This indicates strong agreement between the two approaches (Fig 10).

The deviation between simulation and experiment was below 5% at nearly all measurement points. Specifically, at S2, the deviation ranged from 0.7% to 2%, demonstrating that the simulation accurately represented the central heating behavior. At S1 and S3, deviation ranged from 2% to 5%, mainly due to convective heat loss that was difficult to fully model. Additional sources of deviation included sensor accuracy and the assumption of constant material properties in the simulation.

Despite these differences, the simulation approach proved highly reliable for predicting temperature distribution during induction heating and served as a valuable tool for optimizing the process. Fig 11 presents the temperature variation along Line L at different heating distances (G = 5, 10, and 15 mm), providing insights into heating rate and maximum temperature achieved in the mold.

Fig 11 shows the effect of the heating distance (G) on the heating rate and temperature distribution along Line L. When G = 5 mm, the heating rate was the highest, with the temperature reaching 295 °C within 5 s. This rapid heating is attributed to the small gap between the induction coil and mold surface, which strengthens the magnetic field and increases eddy current density. As a result, heat transfer efficiency improved, and the temperature distribution remained highly uniform with minimal variation between the center and mold edges.

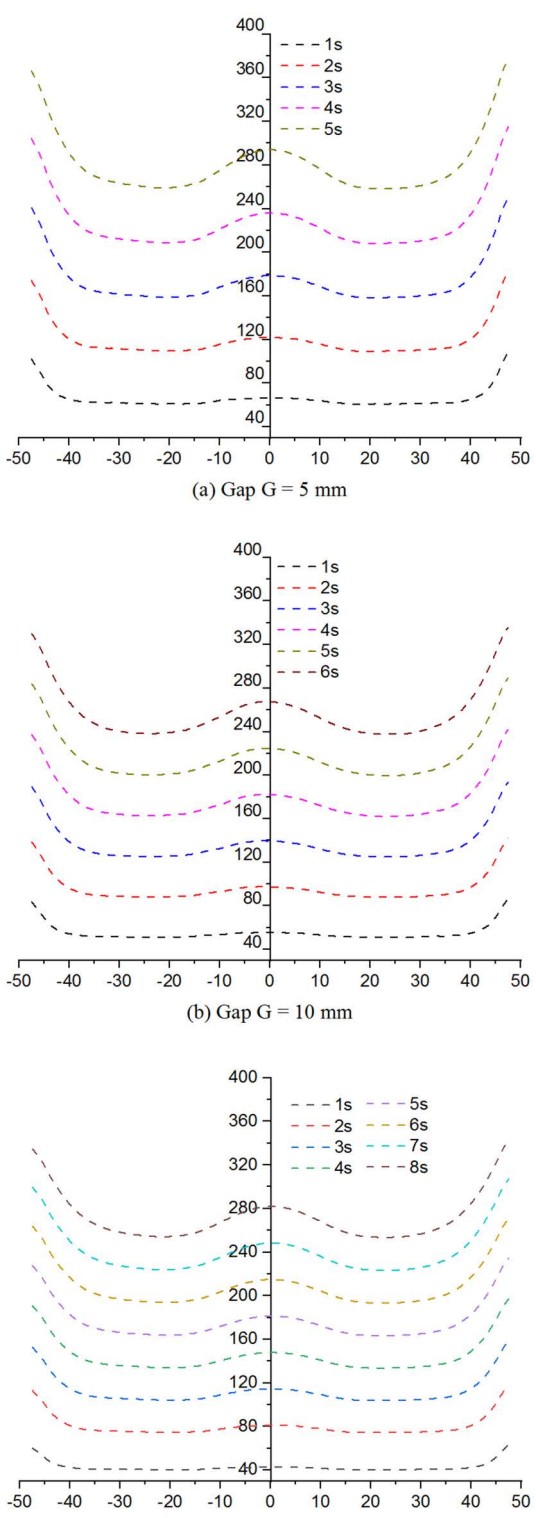

(a) Gap G = 5 mm

(b) Gap G = 10 mm

(c) Gap G = 15 mm

**Fig 11. Time comparison of temperature on Line L with different heating distances (G) by experiment.**

When G increased to 10 mm, the heating rate decreased slightly, reaching 295 °C at approximately 6 s. The larger gap reduced the magnetic field intensity and heat transfer efficiency; however, temperature uniformity remained acceptable, with only ~10 °C difference between the central and edge regions. For G = 15 mm, the heating rate decreased significantly, requiring 8 s to reach 295 °C. At this distance, heat transfer efficiency was low, and the temperature difference between the center and edges increased to 15–20 °C, indicating poor uniformity and potential quality issues during molding.

These results indicate that a smaller G (5 mm) provides the fastest heating and most uniform temperature profile. A G of 10 mm offers a good balance between heating speed and safety, making it suitable for most applications. Although G = 15 mm results in longer heating times, it may be preferred in cases where operational safety or longer cycle times are required. Overall, Fig 11 provides practical guidelines for optimizing the coil-to-mold distance in industrial settings.

The Ex-IHRS system also demonstrated strong agreement between simulation and experiment, with temperature errors below 5% at 295 °C (S2, 5–8 s), outperforming several previous studies. Lucchetta et al. [42] reported ~7% error at 180 °C and required 10 s heating, while Chen et al. [43] achieved 6% error and a 15% tensile strength improvement for PP. Chen et al. [44] reported ~8% error at 200 °C, and Guo et al. [45] achieved 5.5% error with a 10% tensile strength increase for PEEK. In comparison, the Ex-IHRS system not only achieved higher accuracy but also provided more uniform temperature distribution due to its rotational mechanism, making it highly effective for thin-wall injection molding applications

### 3.3. Improvement of the tensile strength of weld lines

In injection molding, weld lines form when two molten polymer flows meet and begin to solidify before completely merging. This phenomenon weakens intermolecular bonding at the convergence point and reduces the tensile strength of the final product. To address this issue, external induction heating was applied to raise the mold surface temperature prior to polymer filling. Localized heating slows the cooling rate of the molten polymer upon contact with the mold, allowing it to remain in a molten state for a longer duration. This extended melting time promotes molecular diffusion and improves bonding at the weld line, leading to enhanced microstructure and superior mechanical performance [36].

Fig 12 shows the injection mold designed for tensile testing, including the sprue, runner, gate, cavity, and thin-wall area. These elements are essential to the molding process, as they directly influence melt flow, weld line formation, and final part quality. The cavity was designed based on tensile testing standards and featured a narrowed central section to focus stress during mechanical testing. This configuration accurately simulates real tensile loading conditions, making it suitable for evaluating the effect of induction heating on weld line strength.

The sprue acted as the main entry point for the molten polymer, directing it into the runner system. The runner distributed the melt evenly to prevent pressure imbalances and ensure consistent filling. The mold included two opposing gates

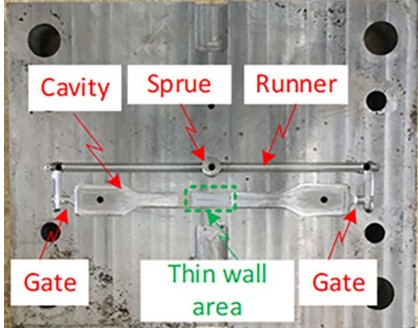

**Fig 12. Tensile sample experiment mold.**

that delivered polymer from both sides of the cavity. As the two melt fronts met at the center, a weld line naturally formed. This region is highly sensitive to poor bonding or air entrapment and is therefore the most critical area for quality improvement. Optimizing mold temperature at this location significantly improves weld line quality.

The thin-wall section simulated challenging industrial conditions where rapid heat loss can cause premature solidification. Without mold heating, this region is prone to defects such as short shots and surface imperfections. Thus, the experimental mold (Fig 12) was specifically designed to replicate real manufacturing scenarios and evaluate the efficacy of induction heating in improving weld line performance. The use of opposing gates ensured the intentional formation of a weld line at the center of the specimen, enabling direct mechanical evaluation. By analyzing the tensile data, this study quantifies the improvement in weld line strength achieved through induction heating, providing an effective solution for manufacturing high-quality injection-molded products.

Increasing mold temperature is a key factor in improving plastic product quality. A higher mold temperature reduces the thermal gradient between the melt and mold surface, minimizing frozen layer formation and enhancing polymer flow, particularly in thin-walled sections. Figs 13 and A4, A5, A6 in S1 Appendix show that for heating distances G = 5, 10, and 15 mm, mold temperature increased with heating time, reducing frozen layer thickness. These figures represent the temperature distribution at mold closure, just before melt filling. This preheating is critical for reducing polymer viscosity, improving flow, and minimizing defects such as weak weld lines.

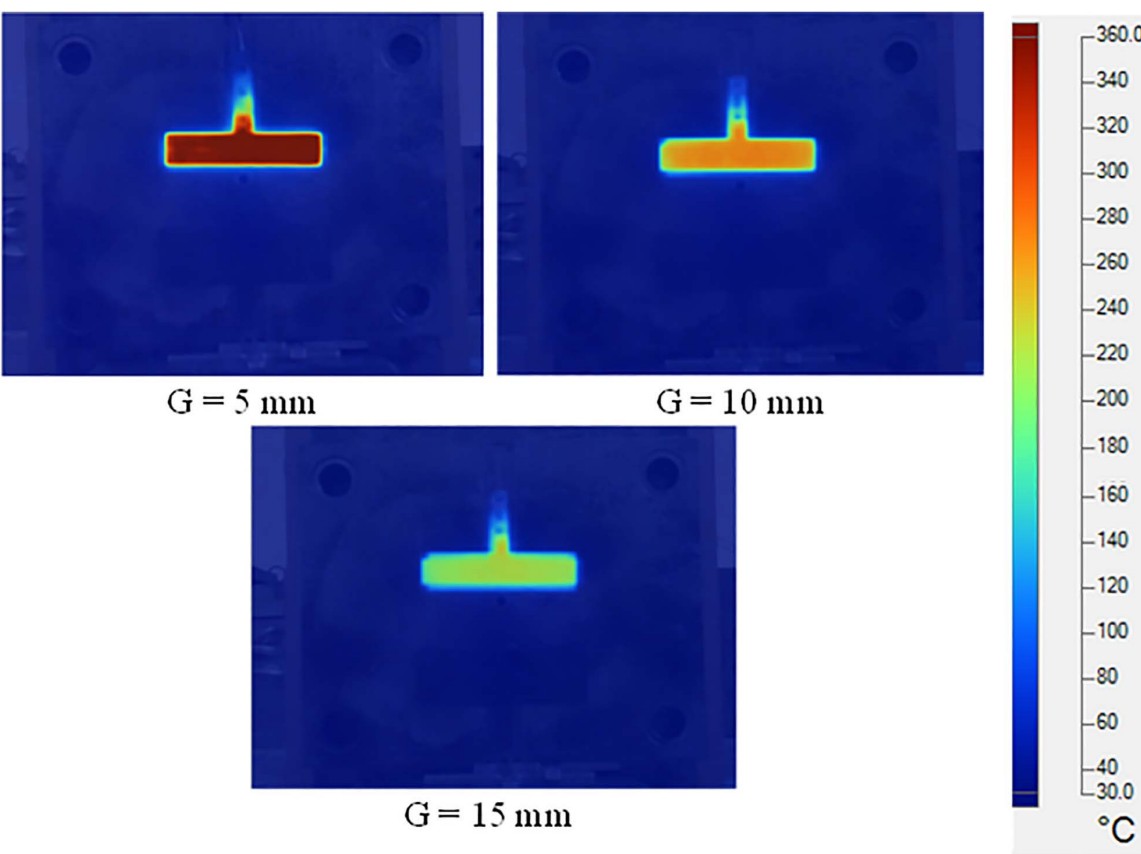

**Fig 13. Temperature distribution of a cavity plate with 5s heating times under a gap (G) of 5, 10, and 15 mm.**

Before each test, the insert was cooled to room temperature using air or water spray and then heated inductively to 295–300 °C at S2 within 5–8 s. At this temperature, any residual moisture evaporated. When mold temperature was low, molten polymer solidified quickly, preventing complete merging and resulting in weak weld lines. At higher temperatures, as shown in Fig. A4, the polymer remained molten upon contact, improving bonding. Figs. A5 and A6 in S1 Appendix show that although G = 10 mm and 15 mm required longer heating times, sufficient temperature was still achieved to enhance weld line quality and reduce weak bonding defects.

At a heating distance of G = 5 mm, the mold heated rapidly and the surface temperature was distributed more uniformly, as shown in Fig 15. After only 5 s of heating, the mold surface temperature reached nearly 300 °C and was maintained during filling, allowing the molten polymer to remain in a fluid state rather than relying solely on the initial insert temperature. This uniform heating reduced weld line defects and improved molecular bonding at the convergence of polymer flows. As a result, the final products exhibited higher tensile strength and fewer defects such as voids or incomplete filling in thin-walled regions. When G = 10 mm, the heating rate decreased slightly, but the target temperature was still achieved, as shown in Fig. A5. After 6 s of heating, the mold surface reached 280–300 °C, which was sufficient to maintain polymer fluidity, especially at the weld line. Although the temperature distribution was slightly less uniform than at G = 5 mm, it still resulted in good weld line quality. The slight increase in heating time remained acceptable for standard production cycles, offering a balanced compromise between heating efficiency and operational safety. At G = 15 mm, the heating time increased significantly, as illustrated in Fig. A6 in S1 Appendix. After 8 s, the mold temperature approached 300 °C, which is adequate to maintain polymer molten state. However, the heating efficiency was lower due to the weaker magnetic field at greater distance. The temperature distribution became uneven, with higher temperatures at the center and cooler edges. This configuration may be more suitable for molding processes with longer cycle times or higher safety requirements rather than rapid production.

Overall, the heating results in Figs 11 and 13 show that a smaller heating distance (G 5 mm) provides the fastest and most uniform heating, making it ideal for thin-walled components and high-quality molding. Nonetheless, G = 10 mm and 15 mm can still achieve the required temperature if heating time is adjusted, making them suitable for longer cycles or when safety is prioritized. These findings provide a practical guideline for optimizing the heating distance and duration in industrial applications.

Fig 14 presents tensile test samples made from PA6, PP, PC, and ABS. These commonly used polymers were selected to evaluate the effect of mold heating on tensile strength and mechanical durability. The results in Fig 15 show that increasing mold temperature significantly improved weld line quality and tensile strength. For PC, the tensile strength increased from 111.9 MPa (no heating) to 123 MPa after 6 s of heating, before decreasing to 112.2 MPa at 8 s due to

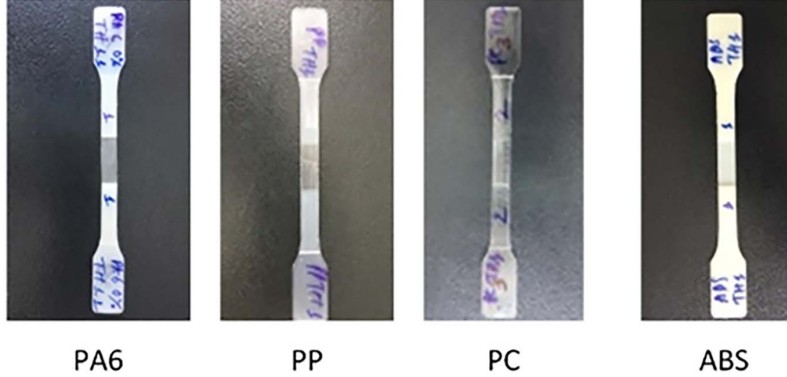

PA6          PP          PC          ABS

**Fig 14. Tensile testing samples with different plastic materials.**

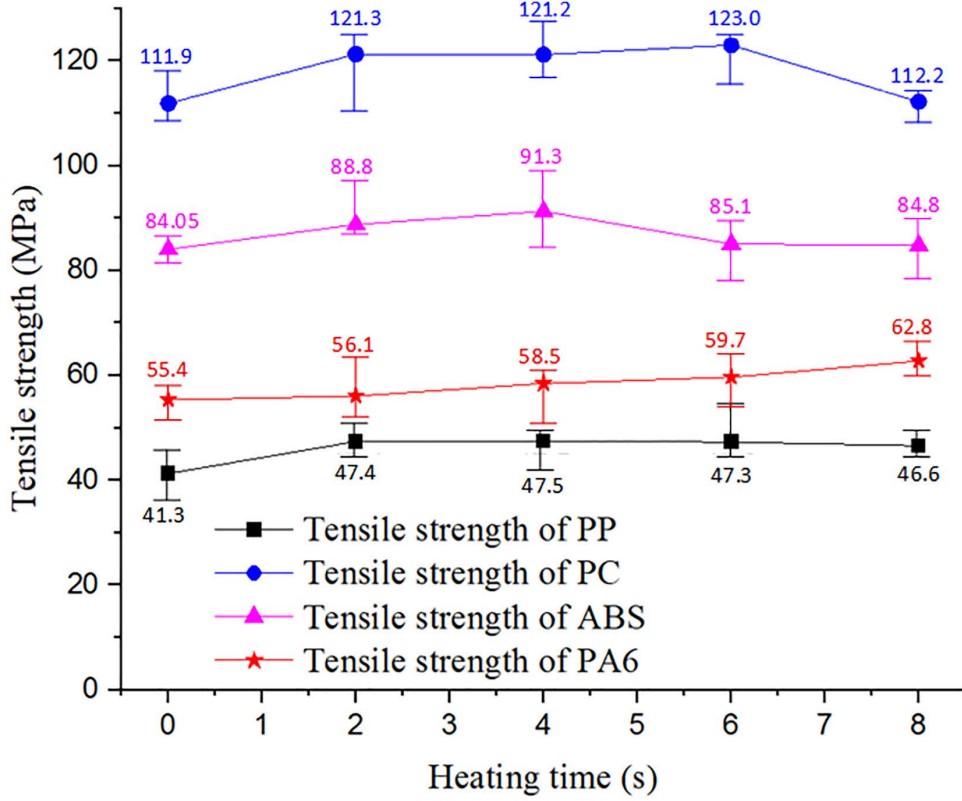

**Fig 15. Effect of the heating time on the tensile strengths of weld-line with different plastic materials under different heating times.**

potential thermal degradation. ABS followed a similar trend, increasing from 84.05 MPa to 91.3 MPa at 4 s, then declining with prolonged heating. PA6 showed continuous improvement from 55.4 MPa to 62.8 MPa at 8 s, while PP increased from 41.3 MPa to 47.3 MPa at 6 s.

These improvements are attributed to higher mold temperatures, which reduce frozen layers, enhance melt flow, and strengthen weld lines. However, excessive heating (beyond 6 s) reduced tensile strength in PC and ABS, likely due to polymer degradation or gas release. Without heating, all materials showed significantly lower tensile strength, confirming that mold heating plays a critical role in mechanical performance. Thus, controlling heating time within the optimal range (4–6 s) is essential for maximizing product quality. Induction heating proves to be an effective method for improving tensile strength and reducing weld line defects, offering a scientific basis for optimizing heating parameters such as distance (G) and duration in injection molding..

## 4. Conclusion

This study proposes a preheating method in which the mold insert is heated externally before being placed into the mold cavity. This strategy shortens the cycle time while maintaining product quality. Both simulation and experimental results confirm that this method provides accurate mold temperature control, reduces the formation of the frozen layer during filling, improves weld line bonding, and significantly enhances the tensile strength of injection-molded products. Experimental data across different polymers demonstrate the effectiveness of this approach. For PC, the tensile strength increased from 111.9 MPa (without heating) to 123 MPa after 6 s of heating. ABS reached a peak tensile strength of 91.3 MPa at 4 s. Similarly, PA6 and PP showed substantial improvements when an appropriate heating duration was applied. However,

excessive heating beyond 6–8 s resulted in decreased tensile strength due to thermal degradation and gas release within the polymer.

The proposed Ex-IHRS system, which uses induction heating with a rotational insert mechanism, outperforms traditional heating methods such as resistance heating, steam heating, and gas-assisted mold temperature control (GMTC). The system rapidly reaches 295 °C at S2 within 5–8 s, with a heating rate of approximately 40 °C/s, compared to GMTC (~28 °C/s) and slower resistance heating. The temperature deviation between simulation and experiment was below 5%, confirming high accuracy and reliable control—critical for thin-wall or microinjection molding. Mechanically, Ex-IHRS increased tensile strength by 14% for ABS (91.3 MPa) and 10% for PC (123 MPa), surpassing GMTC (12% for ABS) and matching dynamic heating results for PEEK. The rotational design maintains temperature stability, improves production efficiency, and reduces cycle time without extending the process. Additionally, this method consumes less energy than full-surface heating. Future work will focus on system optimization, compatibility with advanced composites, and scalability for industrial applications.

## Supporting information

**S1 Appendix.** **Figure A1.** Temperature distribution of Insert at the heating time from 1 s to 5 s with the gap (G) of 5 mm. This is the simulation results. **Figure A2.** Temperature distribution of Insert at the heating time from 1 s to 5 s with the gap (G) of 10 mm. This is the simulation results. **Figure A3.** Temperature distribution of Insert at the heating time from 1 s to 5 s with the gap (G) of 15 mm. This is the simulation results. **Figure A4.** Temperature distribution of a cavity plate with different heating times under a gap (G) of 5 mm. This is the experiment results. **Figure A5.** Temperature distribution of a cavity plate with different heating times under a gap (G) of 10 mm. This is the experiment results. **Figure A6.** Temperature distribution of a cavity plate with different heating times under a gap (G) of 15 mm. This is the experiment results.
(ZIP)

## Acknowledgments

The authors acknowledge the support of HCMC University of Technology and Education for this study.

## Author contributions

**Conceptualization:** Thanh Trung Do.

**Data curation:** Pham Son Minh.

**Formal analysis:** Duc Thuan Huynh, Tran Anh Son, Pham Son Minh.

**Funding acquisition:** Thanh Trung Do, Duc Thuan Huynh.

**Investigation:** Thanh Trung Do, Duc Thuan Huynh, Tran Anh Son.

**Methodology:** Thanh Trung Do.

**Project administration:** Duc Thuan Huynh.

**Resources:** Tran Anh Son.

**Supervision:** Duc Thuan Huynh, Pham Son Minh.

**Validation:** Duc Thuan Huynh, Pham Son Minh.

**Visualization:** Tran Anh Son, Pham Son Minh.

**Writing – original draft:** Thanh Trung Do, Duc Thuan Huynh, Tran Anh Son, Pham Son Minh.

**Writing – review & editing:** Duc Thuan Huynh, Pham Son Minh.

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
