## [Decision Letter · Decision Letter 0]

8 Apr 2025

Dear Dr. Huynh,

We look forward to receiving your revised manuscript.

Kind regards,

Antonio Riveiro Rodríguez, PhD

Academic Editor

PLOS ONE

Journal Requirements:

3. Thank you for stating the following financial disclosure: [This research was funded by the Ministry of Education and Training, project grant No.T2024-02NTĐ, and hosted by Ho Chi Minh City University of Technology and Education, Vietnam.]. 

Reviewers' comments:

Reviewer's Responses to Questions

**Comments to the Author**

1. Is the manuscript technically sound, and do the data support the conclusions?

Reviewer #1: Partly

Reviewer #2: Yes

Reviewer #3: Yes

Reviewer #4: Yes

2. Has the statistical analysis been performed appropriately and rigorously?

Reviewer #1: Yes

Reviewer #2: Yes

Reviewer #3: Yes

Reviewer #4: Yes

3. Have the authors made all data underlying the findings in their manuscript fully available?

Reviewer #1: Yes

Reviewer #2: Yes

Reviewer #3: Yes

Reviewer #4: Yes

4. Is the manuscript presented in an intelligible fashion and written in standard English?

Reviewer #1: Yes

Reviewer #2: Yes

Reviewer #3: Yes

Reviewer #4: Yes

Reviewer #1: The manuscript titled "Applying new mold temperature control strategy to reduce the cycle time and improve the tensile strength of thin wall products in injection molding process" presents an innovative approach to localized heating of the mold cavity using the Ex-IHRSC technique. The study aims to improve part strength and reduce cycle time in the injection molding process. While the concept is intriguing, the manuscript suffers from several critical issues that need to be addressed before it can be considered for publication. Below are detailed comments and concerns:

1) The title claims a reduction in cycle time, but the manuscript lacks concrete experimental data to substantiate this claim. The authors must provide explicit results demonstrating the reduction in cycle time, or the title should be revised to reflect the actual findings.

2) Section 2.1 introduces the principle of induction heating with several equations. However, the values of the parameters used in these equations are not provided, nor are the calculations or results derived from them. This omission makes it difficult to evaluate the validity of the theoretical framework. Please include the parameter values, calculations, and results to support the discussion.

3) The manuscript does not adequately address the potential issue of heat loss when the heated insert contacts the mold. Given the high thermal conductivity of metals, the temperature of the insert is likely to drop rapidly upon contact with the mold. This could significantly affect the efficiency of the heating process and increase energy consumption. The authors should discuss how they have mitigated this issue and provide data on the temperature drop and its impact on energy efficiency.

4) The rotating device for the insert appears to require significant modifications to the mold cavity. This could limit the applicability of the technique to molds with simple geometries and reduce its suitability for complex molds and parts. The authors should discuss the limitations of their technique in terms of mold complexity and propose potential solutions or adaptations for more complex geometries.

5) Figure 4 shows three sensors fixed in the insert for temperature measurement, but the manuscript does not provide a clear image of the insert with the sensors or explain how the sensor cables are managed. This lack of detail raises concerns about the practicality and accuracy of the temperature measurements. Please include a detailed image of the insert with sensors and describe the setup of the cables.

6) The manuscript does not provide data on the temperature of the insert after it is rotated into position for clamping and filling. This is a critical oversight, as the temperature at this stage directly impacts the effectiveness of the heating technique. The authors should include experimental data on the insert temperature after rotation to validate the performance of their technique.

7) The simulation results focus only on the heating process, neglecting the cooling process when the insert is moved away from the induction heater. A comprehensive analysis should include both heating and cooling phases to provide a complete understanding of the thermal dynamics involved. Please supplement the simulation results with an analysis of the cooling process.

8) Figures 15 and 16 illustrate the temperature distribution in the cavity for different heating times, but the manuscript does not specify how the insert temperature was set or how boundary conditions were defined. The authors should carefully describe the boundary conditions and ensure they align with the heating and cooling processes.

9) The simulation appears to separate the induction heating process from the injection filling process, which does not reflect the actual experimental conditions. The authors should clarify whether the insert is heated during the injection process and ensure that the simulation accurately represents the experimental setup.

10) The manuscript only presents tensile strength results for samples made with different polymers. To provide a more comprehensive evaluation of the technique's performance, the authors should include additional mechanical properties such as tensile modulus and failure strain. This will help readers better assess the impact of the insert heating technique on material performance.

11) The manuscript is overly verbose and lacks sufficient effective data to support its conclusions. The authors should focus on presenting clear, concise, and relevant data that directly address the key claims of the study. Redundant or tangential discussions should be removed to improve the manuscript's clarity and impact.

While the proposed Ex-IHRSC technique is promising, the manuscript in its current form does not provide sufficient experimental evidence or theoretical rigor to support its claims. The authors must address the concerns raised above, particularly regarding heat transfer, temperature measurement, simulation accuracy, and experimental data presentation. Once these issues are resolved, the manuscript will be better positioned to make a meaningful contribution to the field of injection molding.

Reviewer #2: Dear Editor and Author:

Thank you for inviting me to review this exciting manuscript.

This study simulated and experimentally investigated the heating process using Ex-IHRSC. The temperature distribution of the insert was investigated by varying the distance between the induction coil and the mold surface (G = 5, 10, and 15 mm) and the heating time (1-8 s). In addition, The effect of the heating time on the tensile strengths of PA6, PP, PC, and ABS was investigated. This emphasizes the practical application of the research. However, the manuscript currently contains numerous errors that need to be fixed. By correcting these issues, the authors can significantly enhance the quality and impact of the work, making it a valuable contribution to the field. Here are the following specific comments and suggestions.(please read the attachment)

Reviewer #3: Dear authors, thank you for the nice publication. Within your text, I added some remarks that can help to make the publication stronger. My main remarks are on the lack of numerical simulation results for the injection moulding process. It is also not clear which Heat transfer coefficient have been used. It should also be mentioned that using induction is consuming a lot of electrical energy.

Reviewer #4: This manuscript studied induction heating of the injection molding mold to improve the weldline strength. An interesting paper, however, I have several comments as follow;

1. Although this idea of induction heating is great, however, must pay attention to the mold insert material if it can sustain the thermal fatigue.

2. How many tensile bars were tested? If more than one, tensile strength should have STD on Figure 19.

3. Unit of viscosity should be Pa*s, density should be kg/m3, 3 is supercript.

4. Can you supply color figures on Figures 8, 9, and 10. What simulation software you used? Can not distinguish the temperature level on your figures. Red color means hot and blue color means cool on color figures.

5. Resolution on Figures 11 and 13 are poor.

6. Caption should be “The weld-line strength”, do you use one gate or two gate? If two gates, must use weld-line strength on Figure 19.

7. Heating time on Figure 12.

**Do you want your identity to be public for this peer review?** For information about this choice, including consent withdrawal, please see our Privacy Policy

Reviewer #1: No

Reviewer #2: No

Reviewer #3: **Yes: ** Prof. dr. ir. Frederik Desplentere

Reviewer #4: **Yes: ** shyh-shin Hwang

---

## [Author Response · Author response to Decision Letter 1]

17 Sep 2025

Dear Reviewers,

We sincerely thank you for your thorough and insightful review of our manuscript. Your expertise and thoughtful comments have been immensely helpful in identifying areas for improvement, allowing us to refine our work with greater precision and clarity. We truly appreciate the time and effort you invested, as it has undoubtedly strengthened the scientific value of our study. In the attached files, we provide a point-by-point response to each of your concerns, detailing the revisions made. All changes are highlighted in the revised manuscript via inline comments for your convenience, with line references to the updated version. We hope these updates address your points effectively and demonstrate our commitment to addressing them in a meaningful way. Your guidance has been invaluable, and we are eager for any additional feedback you may have.

The detail of reply was in the attached file.

Sincerely yours,

---

## [Decision Letter · Decision Letter 1]

5 Oct 2025

Dear Dr. Huynh,

We look forward to receiving your revised manuscript.

Kind regards,

Antonio Riveiro Rodríguez, PhD

Academic Editor

PLOS ONE

Journal Requirements:

Reviewers' comments:

Reviewer's Responses to Questions

**Comments to the Author**

Reviewer #1: All comments have been addressed

Reviewer #4: All comments have been addressed

2. Is the manuscript technically sound, and do the data support the conclusions?

Reviewer #1: Yes

Reviewer #4: Yes

3. Has the statistical analysis been performed appropriately and rigorously?

Reviewer #1: N/A

Reviewer #4: Yes

4. Have the authors made all data underlying the findings in their manuscript fully available?

Reviewer #1: Yes

Reviewer #4: Yes

5. Is the manuscript presented in an intelligible fashion and written in standard English?

Reviewer #1: (No Response)

Reviewer #4: Yes

Reviewer #1: The revised manuscript is much improved and is potentially acceptable for publication. However, the manuscript requires significant condensation before we can proceed with formal acceptance.

The current text is too long and the high number of figures reduces the overall clarity and impact. Please undertake the following revisions:

1) Shorten the Text: Drastically reduce the word count by removing repetitive statements, streamlining the introduction and discussion, and moving detailed methodological descriptions to supplementary information where appropriate.

2) Reduce and Consolidate Figures: The number of figures must be reduced. Please combine related figure panels where possible and move non-essential figures to the supplementary information. Retain only those figures that are absolutely critical for supporting your primary conclusions.

Reviewer #4: Thank you for your revisions, however, next time if you resubmit your revised manuscripts. Please use red color words on your manuscripts or provide another files to show what you have modified to save reviewer’s time. Like ,

Q1: Unit of viscosity should be Pa*s, density should be kg/m3, 3 is supercript.

Reply: Thank you for your comments. Viscosity unit has been modified as Pa*s and density is revised.

**Do you want your identity to be public for this peer review?** For information about this choice, including consent withdrawal, please see our Privacy Policy

Reviewer #1: No

Reviewer #4: **Yes: ** shyh-shin Hwang

---

## [Author Response · Author response to Decision Letter 2]

20 Oct 2025

1. Revisions according to Reviewer #1:

• We have significantly shortened the manuscript by removing repetitive content and streamlining the Introduction and Discussion.

• Detailed methodological descriptions have been moved to Supplementary Information where appropriate.

• We have reduced and consolidated the number of figures, retaining only the essential ones in the main text and relocating non-critical figures to the Supplementary Information.

2. Revisions according to Reviewer #4:

• All unit and notation corrections (e.g., Pa·s, kg/m³ with superscript) have been made.

• We reviewed the entire manuscript to ensure consistency and clarity in formatting, units, and English language usage.

• As suggested, we have clearly highlighted all modifications in the “Revised Manuscript with Track Changes” to facilitate the review process.

---

## [Decision Letter · Decision Letter 2]

17 Nov 2025

Applying a new mold temperature control strategy to improve the tensile strength of thin wall products in injection molding processes

PONE-D-25-07599R2

Dear Dr. Huynh,

We’re pleased to inform you that your manuscript has been judged scientifically suitable for publication and will be formally accepted for publication once it meets all outstanding technical requirements.

Kind regards,

Antonio Riveiro Rodríguez, PhD

Academic Editor

PLOS ONE

Reviewers' comments:

Reviewer's Responses to Questions

**Comments to the Author**

Reviewer #5: All comments have been addressed

2. Is the manuscript technically sound, and do the data support the conclusions?

Reviewer #5: Yes

3. Has the statistical analysis been performed appropriately and rigorously?

Reviewer #5: Yes

4. Have the authors made all data underlying the findings in their manuscript fully available?

Reviewer #5: Yes

5. Is the manuscript presented in an intelligible fashion and written in standard English?

Reviewer #5: Yes

Reviewer #5: Paper can be published in this journal. Paper is well-designed. It is too enough as Technical and scientific.

Experimental is power.

**Do you want your identity to be public for this peer review?** For information about this choice, including consent withdrawal, please see our Privacy Policy

Reviewer #5: No

---

## [Editor Report · Acceptance letter]

PONE-D-25-07599R2

PLOS ONE

Dear Dr. Huynh,

I'm pleased to inform you that your manuscript has been deemed suitable for publication in PLOS ONE. Congratulations! Your manuscript is now being handed over to our production team.

Kind regards,

on behalf of

Dr. Antonio Riveiro Rodríguez

Academic Editor

PLOS ONE